# Identification of Genomic Regions Associated with High Grain Zn Content in Polished Rice Using Genotyping-by-Sequencing (GBS)

**DOI:** 10.3390/plants12010144

**Published:** 2022-12-28

**Authors:** Goparaju Anurag Uttam, Karre Suman, Veerendra Jaldhani, Pulagam Madhu Babu, Durbha Sanjeeva Rao, Raman Meenakshi Sundaram, Chirravuri Naga Neeraja

**Affiliations:** Department of Biotechnology, Crop Improvement Section, ICAR-Indian Institute of Rice Research (IIRR), Hyderabad 500030, India

**Keywords:** biofortification, rice, grain zinc, quantitative trait loci, Genotyping-by-Sequence (GBS), candidate genes

## Abstract

Globally, micronutrient (iron and zinc) enriched rice has been a sustainable and cost-effective solution to overcome malnutrition or hidden hunger. Understanding the genetic basis and identifying the genomic regions for grain zinc (Zn) across diverse genetic backgrounds is an important step to develop biofortified rice varieties. In this case, an RIL population (306 RILs) obtained from a cross between the high-yielding rice variety MTU1010 and the high-zinc rice variety Ranbir Basmati was utilized to pinpoint the genomic region(s) and QTL(s) responsible for grain zinc (Zn) content. A total of 2746 SNP markers spanning a genetic distance of 2445 cM were employed for quantitative trait loci (QTL) analysis, which resulted in the identification of 47 QTLs for mineral (Zn and Fe) and agronomic traits with 3.5–36.0% phenotypic variance explained (PVE) over the seasons. On Chr02, consistent QTLs for grain Zn polished (qZnPR.2.1) and Zn brown (qZnBR.2.2) were identified. On Chr09, two additional reliable QTLs for grain Zn brown (qZnBR.9.1 and qZnBR.9.2) were identified. The major-effect QTLs identified in this study were associated with few key genes related to Zn and Fe transporter activity. The genomic regions, candidate genes, and molecular markers associated with these major QTLs will be useful for genomic-assisted breeding for developing Zn-biofortified varieties.

## 1. Introduction

Micronutrient malnutrition or hidden hunger has become a major issue for people in developing countries, especially in Asia and Africa [1,2]. Marginal intake of micronutrients has been shown to contribute to increasing mortality rates, affect livelihoods, and result in adverse effects on millions of school-going children and pregnant women [3]. About three billion people worldwide are estimated to be affected and more than 24,000 people die daily owing to micronutrient malnutrition or hidden hunger [4]. Recently, the United Nations (UN) declared that tackling micronutrient deficiencies is one of the sustainable development goals (SDGs) and set a goal of SDG 2 to be achieved by 2035 (https://sustainabledevelopment.un.org/sdgs (20 September 2022)). Among different micronutrients, zinc (Zn) and iron (Fe) are the most essential and are associated with the development of physical and mental health in humans and reducing the risk of micronutrient malnutrition [5].

Microelements, such as iron (Fe), zinc (Zn), copper (Cu), and manganese (Mn), are found in many enzymes and have great importance for maintaining normal metabolic pathways in humans. Zn serves as a major co-factor for more than 300 enzymes and 2000 transcription factors and is involved in the metabolism of carbohydrates, lipids, proteins, and nucleic acids in humans [6,7]. It is the only metal present in all six enzyme classes (oxidoreductase, transferase, hydrolases, lyases, isomerases, and ligases) [8]. One third of the human population, particularly children and women, suffer from Zn deficiency-associated health complications such as stunting, diarrhea, reduced immunity, poor cognitive development, and skin problems [9]. According to World Health Organization (WHO), Zn deficiency is sixth among the top 10 major causes of illness in developing nations, affecting 27–30% of the global population. A large portion of the global population suffers from mineral malnutrition, as they depend on plant-based diets that have a low mineral/calorific density [10].

Rice is a predominant and major staple food crop that provides the caloric needs of half of the global population [11,12]. A total of 483.3 million tons of milled rice is produced globally, according to a recent survey (ICRISAT., 2018). To meet the need, however, production must rise to 800 to 900 million tons by 2025 [13]. Mega varieties under cultivation are a poor source of micronutrients, particularly Zn in polished form (12 ± 14 ppm), and provide only one fifth of daily recommended Zn requirements (15 mg/day) [14,15]. In other words, 11 to 15 mg of Zn must be present in 220 gm of rice, which is the per capita amount of rice identified for India. The present international target as well as that from the AICRIP biofortification trial is 28 ppm, which can only meet half of the RDA. Through the AICRIP rice biofortification trial, medium slender, long slender, and short bold grain type rice varieties were released to some of the states in the country. Therefore, there is a need to develop biofortified varieties suitable for rice consumers under aromatic grain and other grain types. To ameliorate this situation, researchers are working to enhance the grain Zn content in rice by different approaches. Biofortification has emerged as a cost-effective and sustainable strategy to tackle mineral deficiencies by enhancing micronutrient content without compromising the yield achieved through agronomic, conventional, and biotechnological approaches [16,17]. The availability of genetic variability in grain mineral nutrients in the rice germplasm can be employed to develop biofortified high-yielding rice varieties [18,19].

Understanding the genetic potential of the genotypes as well as interaction between genotype and environment is essential for developing successful biofortified rice varieties. Developing Zn-biofortified rice is challenging, as it involves complex genetics of the trait, genetic interactions such as epistasis, and environmental factors such as soil and water [14,20,21,22]. Several genetic studies have also been carried out to identify markers linked with quantitative trait loci (QTLs) for high Zn in grains, which would help in the development of biofortified rice varieties through marker-assisted selection (MAS) [23]. QTL mapping provides information for identification of genomic regions associated with targeted traits of interest (Zn) by combining genetic information with phenotypic data [24]. Several QTLs with moderate to high phenotypic variance were reported for grain Zn on all 12 chromosomes of rice [25]. Recent studies on QTL mapping for mineral elements in rice using bi-parental and multi-parental mapping populations have identified multiple loci and demonstrated the genetic complexity of grain micronutrient traits [26].

Compared with other molecular markers (RFLP, RAPD, SSRs, etc.) single nucleotide polymorphisms (SNPs) are preferred because of their uniform distribution and wide occurrence across the genome (one marker every 100–500 bp, specific to species), thus making them the ideal choice for constructing high-density linkage maps and identification of markers closely associated with the trait of interest [27,28]. Next generation sequencing (NGS) technology with reduced cost provides abundant sequence information along with significant improvements in genome coverage and time [29]. Genotyping by sequencing (GBS), an advancement in NGS technology based on genome target reduction and restriction enzyme use has gained popularity as a cost-effective method in the development of genome-wide markers for genetic studies [30].

Numerous methods for reduced representation genome sequencing have been developed and include genotyping by sequencing (GBS) [31], double digest restriction-site-associated DNA sequencing (ddRAD) [32], and specific-locus amplified fragment sequencing (SLAF) [33]. GBS is being deployed for genetic studies in various crops [34,35], particularly GWAS analysis, diversity studies, genomic selection (GS), marker and gene discovery, and high-resolution QTL mapping in rice [36,37,38].

The present study aims to construct a high-density linkage map and identify QTLs for high grain Zn in polished rice and related agronomic traits using a recombinant inbred line (RIL) population derived from a MTU1010 X Ranbir Basmati cross. Putative candidate genes present in the QTLs identified for high grain Zn in polished rice were retrieved. The QTLs and putative candidate genes identified in the study can be deployed to advance genetic research and breeding applications for enhancing grain Zn content in rice.

## 2. Results

### 2.1. Phenotypic Variation in MTU1010 X Ranbir Basmati RIL Population

Wide variations were observed among the RILs across the four seasons for the nine agronomic, yield-, and mineral-related traits, zinc (Zn) in brown rice (ZnBR; 8.5–37.8 ppm), Zn in polished rice (ZnPR; 5.3–31.7 ppm), iron (Fe) in brown rice (FeBR; 1.2–13.3 ppm), Fe in polished rice (FePR; 0.4–6.4 ppm), days to 50% flowering (DFF; 37–131 days), plant height (PH; 8–157 cm), number of tillers per plant (NT; 7–27), panicle length (PL; 16–33 cm), and single plant yield (SPY; 3.6–51 gm) (Table 1). Normal distribution was observed for most of the traits as per the descriptive statistics and histograms across seasons (Appendix A).

In WS 17, G3 RIL showed Zn content above the threshold value of 24 ppm as well as SPY above 26 gm of yield check (IR64) (Figure 1a). In WS 20, G5 RIL showed Zn content above the threshold value (24 ppm) and G3 along with another seven RILs noted SPY. In the DS 18, G1 noted Zn content above the threshold value (24 ppm) and nine lines noted SPY above the yield check (26 gm). In DS 20, G6 noted Zn content above the threshold value and G3 line noted on-par yield with yield check. Based on mean Zn content and SPY of the four seasons, none of the genotypes showed Zn content above the threshold value, whereas seven RILs showed SPY ≥ the yield check (26 gm) (Figure 1b).

### 2.2. Statistical Analysis

#### 2.2.1. Correlation Analysis

The relation between the mean values of individual traits varied from wet season to dry season in the correlation analysis. In the wet season of 2017–2020, positive correlations were observed between SPY and DFF at 0.05 or 0.01 probability. SPY showed significant negative correlation at 0.05 or 0.01 probability. Panicle length showed significant negative correlation at 0.05 probability with FeBR. NT showed significant negative correlation at 0.05 probability with ZnPR. In the dry season of 2018–2020, DFF showed significant negative correlation with FeBR and FePR at 0.001 and 0.05 probability, respectively. Significant positive correlations were observed among the four mineral traits, namely ZnBR, ZnPR, FeBR, and FePR, except for FeBR and FePR in the dry season of 2018–2020 (see Table 2). Significant positive correlations were observed among the four mineral traits ZnBR, ZnPR, FeBR, and FePR, except for FeBR and FePR in the dry season of 2018–2020.

The combined ANOVA across four seasons or environments indicating significant variance at probability < 0.001 was observed among the genotypes (Appendix A).

#### 2.2.2. PCA

To explain the relationships among agronomic, yield-related, and mineral traits, PCA (Principal Component Analysis) with 92 RILs was performed using the mean values of all the four seasons.

The nine traits were resolved into nine principal components (PCs). The first four PCs noted an eigenvalue ≥ 1 (Figure 2) with a cumulative proportion of 75.56% (see Table 2). ZnBR, ZnPR, and FeBR were significant in PC1. PH and PL were negatively significant in PC2; DFF, NT, and SPY were significant in PC3. DFF and NT were significant in PC4 (Figure 2 and Table 3).

### 2.3. Ranking of Genotypes

Ranking of genotypes was conducted using SPY and ZnPR of multi-seasonal data in the “Metan” package of ‘R’. The genotypes G3, G52, G68, G70, G73, G75, and G82 were grouped in WS17 (Figure 3).

### 2.4. High-Density Genetic Map

The genome-wide polymorphic SNPs were evaluated for expected segregation ratio using chi-square analysis in MTU1010 X Ranbir Basmati RIL population. Out of 32,245 SNPs between the parents, a total of 23,183 SNPs with contrasting alleles between the parents were found to be homozygous polymorphic and displayed segregation within the mapping population. A genetic map was constructed using 2746 SNPs spanning a genetic distance of 2445 cM (Table 4). The total number of polymorphic SNPs mapped across 12 chromosomes ranged from 28 (Chr04) to 418 (Chr06), and the average genetic distance ranged from 122 cM (Chr04) to 371 cM (Chr02) (Table 4). A large variation in marker density was seen across the chromosomes, with the highest marker density on Chr12 (2.05 SNPs/cM) and lowest on Chr04 (0.23 SNPs/cM) (Table 4).

### 2.5. QTL Mapping for Agronomic and Mineral Traits

A total of 47 QTLs were identified for eight out of nine traits (except PH). These included 24 QTLs for agronomic and yield traits (11 QTLs for DFF, 4 QTLs for NT, 5 QTLs for PL, and 4 QTLs for SPY) and 23 QTLs for mineral traits (5 QTLs each for ZnBR and ZnPR, 4 QTLs for FeBR, and 9 QTLs for FePR) across four seasons (Table 5 and Table 6). The phenotypic variation explained (PVE) ranged from 3.52% to 36.0%. Altogether, 39 major effect QTLs (PVE ≥ 10%) and 8 minor effect QTLs (PVE < 10%) were identified (Table 5 and Table 6). The highest number of QTLs (16) was observed in the wet season (WS) of 2017.

#### 2.5.1. QTLs for Mineral Traits

##### Grain Zn Brown (ZnBR)

Five major effect QTLs were identified on Chr02 (*qZnBR.2.1*; PVE 20.80%), Chr05 (*qZnBR.5.1*; PVE 23.49%), andChr04 (*qZnBR.4.1*; PVE 16.01%), and one consistent QTL was identified on Chr09 (*qZnBR.9.1* PVE 11.40% and *qZnBR.9.2* 13.88%) in WS20 and DS20 seasons (Table 5). The majority of QTLs showed negative additive value, indicating that the allele contribution was from the donor parent Ranbir Basmati.

##### Grain Zn Polished (ZnPR)

Five QTLs were identified with PVE ranging from 8.10 to 19.42%. Two major-effect consistent QTLs *qZnPR.2.1* (PVE 19.42%) and *qZnPR.2.2* (11.78%) were identified on Chr02 in WS17 and WS20. Further, two QTLs, one major (*qZnPR.6.1*; PVE 12.34%) and one minor (*qZnPR.6.2*; PVE 8.33%), were identified on Chr06. One minor-effect QTL (*qZnPR.7.1*; PVE 8.10%) was also identified on Chr07 (Table 5).

##### Grain Fe Brown (FeBR)

Four major QTLs for FeBR trait were identified on Chr02 (*qFeBR.2.1*; PVE 15.73%), Chr03 (*qFeBR.3.1*; PVE 16.78%), and Chr06 (*qFeBR.6.1*; PVE 11.27%) for WS17 and Chr05 (*qFeBR.5.1*; PVE 34.15%) for WS20. All the QTLs showed negative additive values, indicating that the alleles were contributed from the donor parent (Ranbir Basmati) (Table 5).

##### Grain Fe Polished (FePR)

Nine QTLs were identified for the FePR trait, with the PVE ranging from 3.52% to 36.0%. Three major QTLs on Chr01 (*qFePR.1.1*; PVE 28.73%), Chr05 (*qFePR 5.1*; PVE 11.97%), and Chr06 (*qFePR.6.1*; PVE 12.43%) were identified during WS17. During DS18, four QTLs, two major and two minor, were identified each on Chr02 (*qFePR 2.1*; PVE 3.52%), Chr08 (*qFePR.8.1*; PVE 18.27%), Chr10 (*qFePR.10.1*; PVE 5.53%), and Chr12 (*qFePR.12.1*; PVE 21.55%). One minor QTL was identified on Chr05 (*qFePR.5.2*; PVE 8.68%) for WS20 and one major QTL was identified on Chr03 (*qFePR.3.1*; PVE 36.0%) for DS20 (Table 5).

#### 2.5.2. QTLs for Agronomic and Yield-Related Traits

##### Days to 50% Flowering (DFF)

A total of nine QTLs were identified for the DFF trait across four seasons. The phenotypic variance (PVE) ranged from 6.63% to 21.35%. One QTL each was observed on Chr01 (*qDFF 1.1*, PVE 8.86%), Chr03 (*qDFF.3.1*, PVE 9.43%), Chr08 (*qDFF.8.1*, PVE 20.96%), and Chr11 (*qDFF.11.1*, PVE 19.25%).

Three major effect QTLs, *qDFF.2.1* with PVE 16.05%, were observed in WS17, and another two consistent QTLs (*qDFF.2.2*, PVE 10.32% and 12.25%; and *qDFF.2.3*, PVE 14.40% and 14.31%) were observed in DS18 and DS20, respectively. On Chr06, one major (*qDFF.6.2*; PVE 21.35%) and one minor effect (*qDFF.6.1*; PVE 6.63%) QTLs were identified (Table 6).

##### Number of Tillers (NT)

Four major QTLs, two QTLs on Chr09 *qNT.9.1* (PVE 17.94%) and *qNT.9.2* (PVE 10.37%) were identified. The additive effect of these two QTLs indicates inheritance from a donor parent (Ranbir Basmati). Furthermore, two QTLs each on Chr06, *qNT.6.1* (PVE 12.03%) and Chr10, *qNT.10.1* (PVE 11.02%) were identified (Table 6).

##### Panicle Length (PL)

Four major QTLs for panicle length (PL) were identified on Chr02 (*qPL.2.1*) with PVE 13.29% in WS17. Two major QTLs on Chr11, *qPL.11.1* (PVE 12.39%) and *qPL.11.2* (PVE 16.51%) were identified for DS18 and DS20. Further, one major effect QTL on Chr12, *qPL.12.1* with PVE 15.88% was identified for WS20 and one minor effect QTL *qPL.9.1* with PVE 8.03% on Chr09 was observed (Table 6).

##### Single Plant Yield (SPY)

Four major QTLs were identified for three seasons, with the PVE ranging from 11.26% to 15.32%. Two QTLs, namely *qSPY.9.1* (PVE 11.28%) and *qSPY.9.2* (PVE 12.99%), were identified on Chr09. Another two QTLs each on Chr03 (*qSPY.3.1*; PVE 11.26%) and Chr06 (*qSPY.6.1*; PVE 15.32%) were identified with an additive effect value of −4.98, indicating the allele inheritance from donor parent Ranbir Basmati (Table 6).

### 2.6. Co-Localization of QTLs

In our present study, the co-localization of QTLs for traits was observed in the genic regions of Chr02, 05, and 06. DFF (days to 50% flowering) on Chr02 (SNP_ 24248265–SNP_ 10686645) was co-localized with four mineral trait QTLs: FeBR, FePR, ZnBR, and ZnPR. Similarly, the region on Chr05 (SNP_14288078–SNP_24423261) FePR QTL was co-localized with ZnBR QTL (Table 5, Table 6 and Figure 4).

#### 2.6.1. Consistent QTLs across Seasons

Two major consistent QTLs, *qDFF.2.2* and *qDFF.2.3,* were identified across two seasons (DS18 and DS20), and four major mineral QTLs, *qZnBR.2.1, qZnPR.2.2* on Chr02, were consistent in WS17 and WS20. Another two QTLs, *qZnBR.9.1* and *qZnBR.9.2* on Chr09, were consistent in DS20 and WS20 (Table 5, Table 6 and Figure 4).

#### 2.6.2. Common QTLs

Two common QTLs for grain Zn were identified on Chr02. The QTLs for grain Zn polished, *qZnPR.2.1* (PVE 19.42%), and grain Zn brown, *qZnBR.2.1* (PVE 20.80%), on Chr02 were located in the 25.68–24.25 Mb region (Table 5, Table 6 and Figure 4).

### 2.7. Analysis of Epistatic Interactions

A total of 170 epistatic interactions were detected for agronomic and yield (83) along with grain mineral (87) traits across four seasons (Appendix A). Eight epistatic interactions for five traits (PH, NT, FeBR, FePR, and ZnPR) were considered as the main effect QTLs, with PVE ranging from 11.20 to 26.83%. For two QTLs of ZnPR, one on Chr05 interacted with Chr10 (PVE 12.15%) and another QTL on Chr07 interacted with Chr11 (PVE 17.24%). Epistatic interaction for FeBR trait was observed on Chr04, which interacted with the locus on Chr05 (PVE 16.16%). Similarly, another epistatic interaction was observed for FePR on Chr02, which interacted with locus on Chr10 (PVE 26.83%). Two interactions were observed for NT on Chr01 with two loci of Chr02 and Chr08, and another QTL for NT on Chr02 interacted with loci on Chr05. Two digenic epistatic QTL interactions within the two loci of Chr02 were observed for PH, with PVE 8.74% and 24.49%, respectively (Figure 5).

### 2.8. Candidate Genes Underlying QTLs for Zn QTLs

A total of 1637 genes present within the identified QTL region for Zn brown (ZnBR) and polished (ZnPR) on Chr02, Chr06, and Chr09 were retrieved (Appendix A). They were annotated and categorized broadly on cellular, molecular, and biological processes using WEGO (Web Gene Ontology Plot) software (Appendix A). Among several functional categories, 36 putative candidate genes (seven on Chr02, 18 on Chr06, and 11 on Chr09) belonging to transporter activity roles were shortlisted for the targeted QTLs (Appendix A). Out of 36 genes, two candidate genes on Chr06 showed a putative function of Zn and Fe transport; thus, the role of these candidate genes in Zn transport and metabolism was evaluated using knetminer software.

Os06g0705700 present in the QTL region of qZnPR.6.2 encoded by peptide transporter was found to be linked with 17 genes and three QTLs. Knetminer network analysis showed that this gene has a role in Zn and Fe transport (Figure 6a), and another gene Os06g0706100 present in the same QTL region on Chr06 is encoded by NRAMP4 and has a role in metal ion transport, particularly Zn transport. The two putative candidate genes (Os06g0705700 and Os06g0706100) within the identified QTL region are being further investigated (Figure 6b).

### 2.9. Expression Analysis of Identified Candidate Genes

To predict the expression pattern of selected Zn- and Fe-related candidate genes (Os06g0705700 and Os06g0706100), expression profiling was carried out using the two parental lines MTU1010 and Ranbir Basmati, their derived inbred lines (I-42, I-44, and I-83), two mega rice varieties (IR64, BPT5204), a set of Zn-biofortified varieties (ZINCORICE, PAUSTIC1, PAUSTIC9) and a high-Zn landrace karuppunel. The recipient parent MTU1010, which has low Zn content, showed low expression of both the genes, whereas Ranbir Basmati showed higher expression levels (Appendix A).

## 3. Discussion

One third of the global population is reported to be suffering from a lack of sufficient Zn nutrition [39]. Biofortification is considered as long-term sustainable strategy to address micronutrient malnutrition by enhancing the grain micronutrient density. Over the last decade, using conventional breeding approaches, a few biofortified rice varieties with high grain Zn have been developed and released globally, as Zn plays a critical role in several cellular and metabolic activities [40,41]. Though wide variability for grain Fe is available in brown rice, because of polishing, around 70 to 80% of grain Fe is lost [42].

Marker-assisted breeding has a vast potential to achieve desirable phenotypic variations in less time through the deployment of molecular markers linked to QTLs for desirable traits [43]. Large QTL regions identified with low-density SSR markers may have undesirable linkages, resulting in unsuccessful introgression [44]. In some of the recent studies, undesirable linkage groups were successfully eliminated by employing NGS-generated markers such as SNPs by identifying the recombinants. Next generation sequencing (NGS) technologies have become potential tools for the discovery of millions of SNPs in a cost-effective manner. The genotyping-by-sequencing (GBS) technique has facilitated the identification of key genomic regions for both complex and simple traits and has accelerated the breeding process [45,46,47,48]. ddRAD sequencing facilitates the identification of SNPs by reducing genome complexity in rice and other crops [49].

In the present study, we developed a high-density genetic linkage map from an RIL population MTU1010 X Ranbir Basmati using a GBS approach to identify QTLs for grain Zn and Fe using multi-seasonal phenotype data of the mapping population. Detailed analysis of phenotype data revealed a wide genetic variability within the RIL population for grain Zn and Fe along with agronomic and yield-related traits and showed normal distribution across four seasons. Previous studies indicated that complex inherited traits and micronutrient values exhibit wide variation, and these can be attributed to various genetic and environmental factors [50,51]. We examined data from the two wet and dry seasons in the current study and confirmed our results based on the previous variability studies in rice grain mineral content across the seasons (Kharif and Rabi) [52,53].

### 3.1. Correlation

Highly significant positive correlations were obtained among ZnBR, ZnPR, FeBR, and FePR, irrespective of season and location; however, there was a negative correlation with single plant yield (SPY). Several studies have reported positive correlation between Zn and Fe in rice [54,55,56,57]. Swamy et al. (2018b) [58] attributed the positive correlation between Zn and Fe contents to commonalities in their pathways and genetic networks of Zn and Fe uptake, translocation, and loading in rice. Negative correlations between yield and Zn in rice were earlier reported by [20,59] (G3: Mean Zn and Mean SPY). A QTL (*q*GZn9a) identified in Australian wild rice strain *O.meridionalis* was reported to be associated with an increase in grain Zn levels also found to be concomitant with fertility reduction [60]. The negative linkages between the yield and Zn must be eliminated for developing a successful biofortified variety by generating a greater number of recombinants per cross for identifying positive combinations for grain Zn and yield.

### 3.2. Identification of Promising RILs

Six stable RILs were identified for four mineral traits in the present study with >24 ppm (Zn) based on stability and G × E interaction analysis across four environments. From the study, we could also identify promising RILs with high grain Zn in polished rice (35.6 ppm) and promising RILs with single plant yield (28.4 g to 38.4 g), which could be further deployed in the breeding of biofortified rice varieties. For rice grain Zn and Fe, stability and G × E analyses are used for identification of stable donors [47,61,62,63].

Considering the wide variability observed for the breeding lines with high grain Zn and Fe, stability and G × E analyses are being applied for selecting promising RILs in cereals. The contribution of environmental variation to grain Zn and Fe along with other agronomic traits in a RIL population of sorghum was demonstrated through genotype · environment interaction, correlation, and GGE biplot analyses [64]. Stable RILs with higher grain Zn and Fe content were also identified in the RILs of pearl millet using AMMI and GGE biplot analyses [65]. Different stable breeding lines were identified for different environments among eight Zn biofortified lines through stability and G × E analyses [51].

### 3.3. Identification of Quantitative Trait Loci (QTLs)

Out of 13 QTLs identified for grain Fe and 10 QTLs for grain Zn in brown and polished rice using ICIM, six QTLs were found to be stable/consistent across the seasons, namely *qDFF.2.2*, *qDFF.2.3*, *qZnPR.2.1*, *qZnPR.2.2*, *qZnBR.9.1*, and *qZnBR.9.2* on Chr02 and Chr09. The lack of stability of QTL effects across the environments/genetic backgrounds has been one of the most limiting factors for successful deployment of QTLs through MAS breeding for various complex traits [66,67,68,69]. The consistent QTLs (*qZnPR.2.1*, *qZnPR.2.2*) identified in this study can reliably be deployed in marker-assisted breeding for high grain Zn.

Several researchers have mapped QTLs for grain Zn and Fe traits in rice using various mapping populations, such as RILs, ILs, F2, DH, and MAGIC populations, and have identified multiple loci and clearly demonstrated the genetic complexity of the grain micronutrient traits. The genomic region on Chr02 (SNP_24248265-SNP_25682947) has shown co-localization in the *qZPR.2.1* region [42].

Co-localization of QTLs for different element contents in grain is common in rice [70]. In the study, co-localization was observed: the genomic regions of Chr02, 05, and 06 of the DFF trait was co-localized with mineral trait QTLs FeBR, FePR, ZnBR, and ZnPR on Chr02. The Fe QTLs on Chr05 were co-located with ZnBR QTL, suggesting that there may be a possibility of selecting high grain Zn and Fe lines using molecular markers; Ref. [71] reported co-location of Zn and Fe QTLs on Chr02, 03, 08, and 12 in RIL populations.

In the present study, eight significant epistatic interactions were observed for PH, NT, FeBR, FePR, and ZnPR on Chr01, 02, 04, 05, and 07 and accounted for 11.20–26.83% PVE, suggesting the complex genetic variation of the traits. Two digenic interactions were detected for PH on Chr02, with PVE 8.74% and 24.49%, respectively. None of the identified digenic interactions in the present study were found to be involved with the main effect QTLs. Similar observations were earlier reported for epistatic interactions for grain Zn in rice [52,59]. Recent QTL mapping studies suggest that epistasis is considered to be a crucial genetic component underlying complex quantitative traits. Epistasis should be underscored in studying complex traits because it can account for hidden quantitative genetic variations in natural populations [72].

Integration of genomics with conventional breeding efforts can decipher the molecular mechanisms underlying traits of interest. Using the rice genome sequence, it is now possible to identify the putative candidate genes for grain Zn within QTL regions. In the present study, major QTLs involved in grain mineral element content that show consistency across seasons merit further examination by fine mapping and candidate gene analysis for ultimate use in MAS. The identified five candidate genes Os02t0629200 on Chr02 with Zn binding and transport activity, two genes Os09t0268300 and Os09t0297300 on Chr09 with transporter activity, and two genes Os06t0705700 and Os06t0706100 on Chr06 with Zn and Fe transporter activity and metal ion transport function could be promising genes for further validation. The enhanced expression of Os06t0705700 and Os06t0706100 in the flag leaf samples of the donor parent (Ranbir Basmati) and some of the donors/RILs with grain Zn further supported the identified QTLs/candidate genes in the present study.

Out of 92 RILs evaluated, one promising RIL G3 showed the highest SPY and Zn content across 92 RILs. The G3 RIL can be used as donor in biofortification breeding programs. Likewise, three other RILs, G1, G4 and G6, with high Zn content donors, and RILs G73, G49, G70, G82, and G52, with promising yield, can be used for future breeding programs.

## 4. Materials and Methods

### 4.1. Development of RIL Population

A recombinant inbred line (RIL) population (*n* = 306) was developed using the single seed descent (SSD) method by crossing MTU1010 (also known as Cotton Dora Sannalu, pedigree: Krishnaveni/IR-64; Zn: 16.6 ppm in brown rice; 12.9 ppm in polished rice; mean yield of 6.5 tons ha^−1^; widely cultivated mega variety) as a recipient parent for grain zinc (Zn) and Ranbir Basmati (pure line selection from Basmati 370-90-95; mean yield of 2–2.5 tons ha^−1^; Zn: 27.5 ppm in brown rice; 23.4 ppm in polished rice) as a donor parent. In this study, two parental lines and a set of selected 92 RILs (F_8_) were used for genotyping and phenotypic analysis. All 306 lines’ Fe and Zn values are included in Appendix A.

### 4.2. Field Experiment Details

A total of 92 RILs along with parents were evaluated during four seasons (Wet season (WS) 2017; Dry season (DS) 2018; Wet season (WS) 2020 and Dry Season (DS) 2020) at ICAR-Indian Institute of Rice Research (IIRR) farm, Hyderabad, India (17.53° N latitude and 78.27° E longitude with 545 mm mean rainfall). Each RIL was planted in four rows. with each row consisting of 15 plants with a spacing of 20 × 15 cm following randomized block design (RCBD). The crop was raised by following recommended field management practices of production and protection.

### 4.3. Phenotypic Evaluation of Agronomic and Mineral Traits

Across four environments (seasons), five uniform plants were tagged from the center of each row, and phenotype data was collected for the traits. Days to 50% flowering (DFF), plant height (PH) (cm), number of tillers per plant (NT), panicle length (PL) (cm) at heading/early filling stage, single plant yield (SPY) (g), grain zinc (ZnBR, ZnPR), and iron (FeBR, FePR) in brown and polished rice were estimated at the post-harvest stage.

### 4.4. Estimation of Grain Zn and Fe Content

For estimation of grain Zn and Fe in brown rice (ZnBR, FeBR) and polished rice (ZnPR, FePR), seed samples were dehusked using a JLGJ4.5 rice husker (Jingjian Huayuan International Trade Co., Ltd., Hangzhou, Zhejiang, China). The brown rice was polished using a polisher with non-ferrous and non-zinc components (Krishi International India Ltd., New Delhi, India). After a thorough cleaning, samples of brown and polished rice (5 g) were subjected to energy-dispersive XRF (ED-XRF) (OXFORD Instruments X-Supreme 8000, Abingdon, UK) as per the standardized protocol [73].

### 4.5. Statistical Analysis

For each season, descriptive statistics such as mean, standard error of the mean (SEM), skewness, kurtosis, and coefficient of variations (CV%) were calculated for five plants using MS-Excel (2010). Trait-wise frequency distribution histograms were generated using R software [74]. Pearson correlation analysis and principal component analysis were computed using R script among nine traits. Different R packages (R core Team, 2018, Vienna, Austria), namely ggplot2, gge, FactoMinerR, and factoextra, were used to generate frequency distribution plots and individual comparison of RILs; the ‘Metan’ package was used for ranking of RILs [75].

### 4.6. Genotyping by Sequencing (ddRAD-Seq)

Genomic DNA was extracted from pooled young leaves (20–25 days) of both parents and RILs using the DNeasy plant kit (Qiagen, Hilden, Germany) following the manufacturer’s protocol. DNA quality of each sample was checked on 0.8% agarose gel. Furthermore, DNA quantification was done using a Qubit 2.0 fluorometer (Themo Fisher Scientific Inc., Waltham, MA, USA). ddRAD–Seq protocol (modified GBS) was followed in the present study.

To perform GBS, genomic DNA was double digested with *SphI* and *MlucI* restriction enzymes (NEB, UK) and fractionated in 2% agarose gel to check the product size. The digested fragments were cleaned (Agencourt AMpure XP beads, Invitrogen, Waltham, MA, USA) using standard protocols. The ligation enzyme, T4 DNA ligase (NEB, England) was used to ligate the unique barcode adapters (4–8 nt sequence) at 16 °C for 30 min and heat-inactivated at 65 °C for 10 min.

This was followed by indexing with the addition of index-1 and index-2 (6–8 nt long) for multiplexing sequencing libraries in NGS Illumina. These libraries were PCR amplified (8–12 cycles) using a Phusion^TM^ polymerase kit (Fisher Scientific, Loughborough, UK) followed by AMPure bead cleanup for purification to remove excess adapters and were sequenced on the Illumina HiSeqX platform (Illumina Inc, San Diego, CA, USA).

#### 4.6.1. Genotyping and Filtration

The sequence reads for the parents and RILs were obtained as FASTQ files, which are used for SNP discovery. Raw reads were de-multiplexed according to their barcodes, and the adapters/barcode sequences were removed using standard software [76]. High-quality reads were aligned onto the rice reference genome of Nipponbare (MSU7) using bowtie2-2.2.6. The mapped reads were exported in the form of a Sequence Alignment Map (SAM) file by SAM tools, version 0.1.19 [77]. The alignment file was then processed for SNP calling using a Bayesian approach at specific site. Potential SNPs were filtered using the following criteria: loci with >70% missing data and those that showed distorted segregation of the two parental genotypes were excluded. SNPs with minimum allele frequency (MAF) > 0.05 and 90% call rate were considered for further analysis. The variant annotation was performed based on rice gene models, using in-house pipeline software developed by Agri Genome Labs Pvt., Ltd., Kochi, India.

#### 4.6.2. Genetic Linkage Map Construction

High-quality SNPs obtained after filtering were used for map construction using the linkage mapping function using IciMapping v4.2. [78]. The grouping and ordering of 2746 SNP events between adjacent markers was performed at a minimum logarithm of odds (LOD) value of 2.5. ‘Rippling’ was done for fine-tuning of the ordered markers on their respective chromosomes. The Kosambi map function was used for the construction of the genetic map and to convert the recombination frequencies into map distances in centiMorgans (cM) [79].

#### 4.6.3. QTL Analysis

The main effect QTL analysis was carried out using IciMapping. Deploying the BIP function in IciMapping, additive QTLs were identified by inclusive composite interval mapping (ICIM) based on a 1000—permutation test at a 95% confidence level. The QTLs with >3.0 LOD and phenotypic variance explained (PVE) > 10% were considered as major effect QTLs, those with PVE <10% were considered as minor effect QTLs for a particular trait. Epistatic interactions with the logarithm of odds (LOD) threshold value at 5.0 were considered as significant epistatic QTLs. The QTLs were visualized using MapChart V2.3 [80].

### 4.7. Mining of Candidate Genes

To identify the candidate genes within the identified QTL region, the genomic position of the flanking markers was used. Using the rice genome annotation project ((http://rice.plantbiology.msu.edu/ (22 August 2022)), RAP-DB genome browser (https://rapdb.dna.affrc.go.jp (22 August 2022)), and Q-TARO (QTL Annotation Rice Online) database (Yonemaru et al., 2010) [81] (http://qtaro.abr.affrc.go.jp/ (22 August 2022)), the candidate genes within the QTL regions were retrieved. Genes were functionally characterized into various categories using WEGO [82]. Network analysis of genes present in the major QTL genomic regions was derived using the Knetminer program (http://knetminer.rothamsted.ac.uk/Oryza_sativa/ (24 August 2022)). The temporal and spatial expression of the identified candidate genes were studied using RiceXpro (www.ricexpro.dna.affrc.go.jp (24 August 2022)).

### 4.8. Confirmation of Identified QTLs and Markers

Out of mined candidates from QTLs, primers were designed for two candidate genes with known function of Zn transport, namely *Os06g0705700* (CS528–F primer: GTGACGGCTTCGGATGAG and R primer: CCCGGTGTAGAAGAAGGTAATG), *Os06g0706100* (CS533–F primer: CCAATGCCTGGCCTACTT and R primer: CCACGTGGACACGTTCTT) using the Primer Quest tool (https://eu.idtdna.com/pages/tools/primerquest (2 September 2022)) with qPCR parameters (amplicon size < 150 bp; GC content > 50%; melting temp ~60 °C; no secondary structures) and were synthesized at Integrated DNA Technologies (USA). Based on the RiceXpro data, flag leaf samples during the anthesis stage (after 10 days of flowering) from 11 genotypes (with differential grain Zn) were collected in RNALater™ (Sigma Aldrich, St. Louis, MO, USA). The published protocols for RNA isolation, cDNA synthesis, and checking of integrity of RNA and cDNA and their concentration were followed [83]. To investigate the expression of *Os06g0705700* and *Os06g0706100* genes, qPCR was performed with three biological replicates. Each qPCR reaction was performed in three technical replicates containing 5 µL SYBR^®^ qPCR Master Mix (Promega Corporation, Madison, WI, USA), 0.6 µM forward and reverse gene-specific primers, and 100 ng template cDNA. The qPCR was performed using QuantStudio5 Applied Biosystems (Thermo Fisher Scientific, Waltham, MA, USA) Real-Time PCR Detection System with the following conditions: 40 PCR cycles of denaturation at 95 °C for 15 s, amplification at 58 °C for 15 s, followed by extension at 72 °C for 10 s. The melting curve (melting—95 °C for 10 s, 65 °C for 1 min, 97 °C for 1 s) was analyzed for checking the amplicon specificity. Two internal control genes, Membrane Protein (Memp) (LOC_Os12g32950.1-F Primer: GAGCGC AAAGTTCCAGAAGAA and R Primer: CGCCACTAGTTGCCGTCCTGAT) and Tumor Protein Homolog (TPH) (LOC_Os11g43900.1-F Primer: CATTGGTGCCAACCCATC and R Primer: AAGGAGGTTGCTCCTGAAGA) were used for the normalization [84]. Relative fold change was calculated using the 2(−ΔΔC(T)) method as described by [85].

## 5. Conclusions

The developed genetic map in the present study using MTU1010 X Ranbir Basmati cloud be a foundation for fine mapping of grain zinc (Zn) QTLs, and the identified genomic regions could be utilized in future breeding programs. The genes present in the identified QTL regions in this study reportedly play a key role in transporter activity. Further fine mapping, sequencing, and functional validation is required to identify the candidate genes in these QTL regions.

## Figures and Tables

**Figure 1 plants-12-00144-f001:**
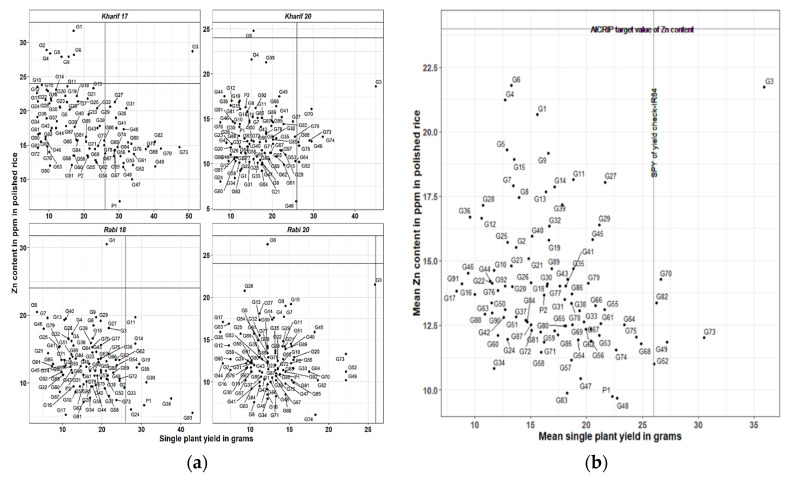
(**a**) Individual comparison of 92 RILs with respect to zinc (Zn) content and single plant yield (SPY) in four seasons; (**b**) mean zinc (Zn) content vs. mean single plant yield (SPY) across four seasons.

**Figure 2 plants-12-00144-f002:**
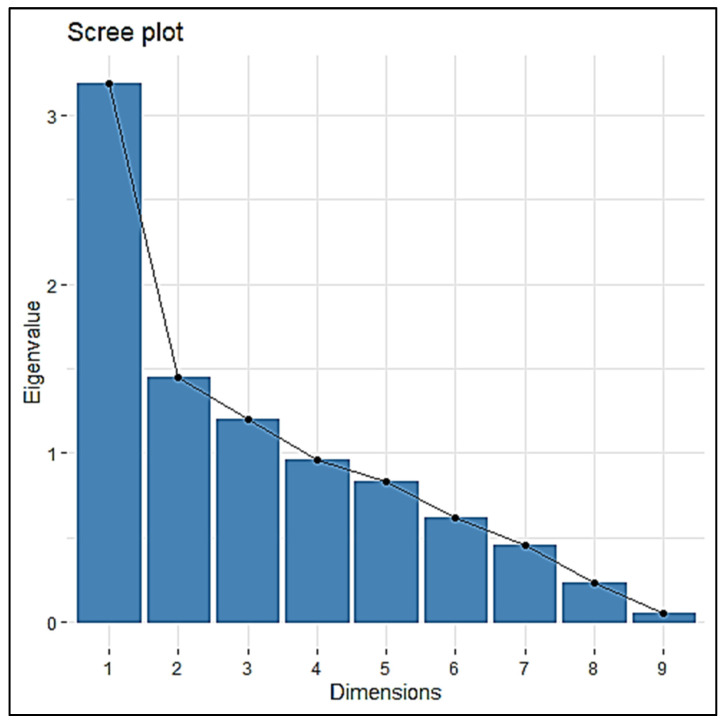
PCA of 92 RILs for nine traits based on mineral, yield, and agronomic traits.

**Figure 3 plants-12-00144-f003:**
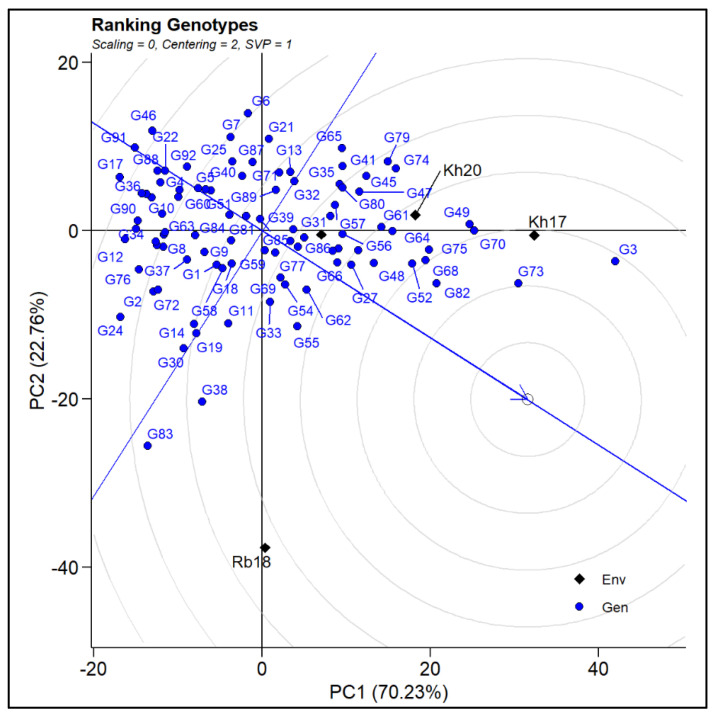
Ranking of genotypes across all seasons.

**Figure 4 plants-12-00144-f004:**
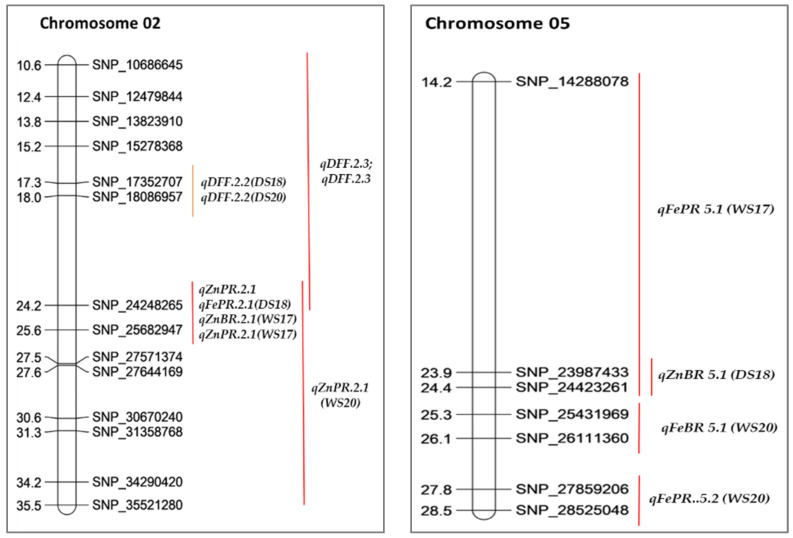
Major quantitative trait loci (QTLs) for various mineral, agronomic, and yield-related traits identified in the MTU1010 X Ranbir Basmati RIL population.

**Figure 5 plants-12-00144-f005:**
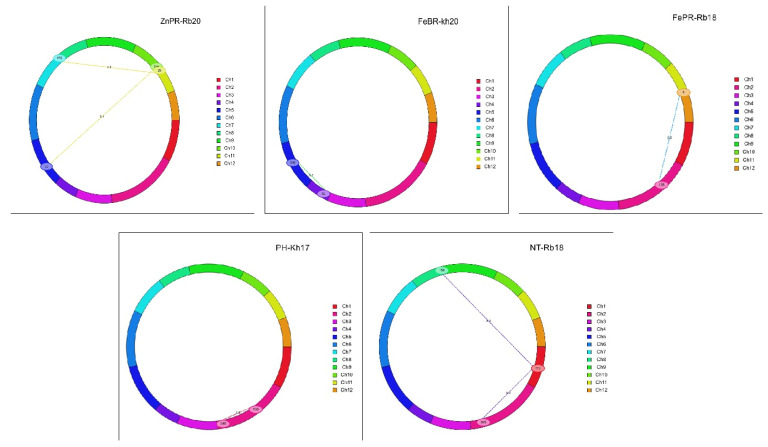
Epistatic interactions in MTU1010 X Ranbir Basmati RIL population using IciMapping v4.2.

**Figure 6 plants-12-00144-f006:**
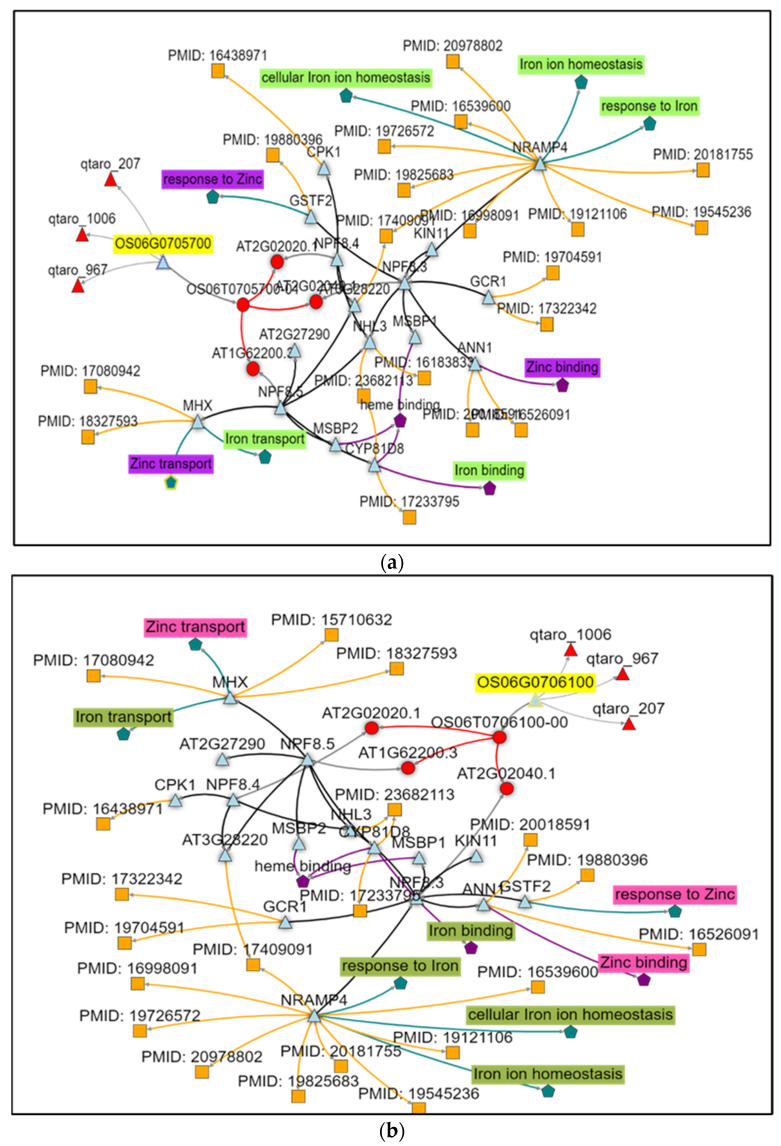
(**a**) Network analysis of candidate gene (Os06g0705700) in a major QTL *qZnPR.6.2* using Knetminer analysis; (**b**) Network analysis of candidate gene (Os06g0706100) in a major QTL *qZnPR.6.2* using Knetminer analysis.

**Table 1 plants-12-00144-t001:** Descriptive statistics of selected RILs (four seasons).

Statistics	Year	DFF	PH	NT	PL	SPY	ZnBR	ZnPR	FeBR	FePR
Mean	WS17	105.9	126.6	11.2	26.5	20.6	22.2	18	9.1	2.8
	MTU1010	97	92	12	21.1	30.21	12.4	6.8	10.1	1.9
	R. Basmati	92	115	18	19.7	21.1	17.2	13.3	11.6	2.9
	DS18	99.4	110.8	14.6	22.4	17.4	16.7	13	9.4	2
	MTU1010	96	90	13	22.2	30.7	11.7	6.7	10.1	1.9
	R. Basmati	85	114	19	19.4	14.6	16.6	12.5	11.6	2.9
	WS20	104.6	119.1	11.9	25.6	17.2	17.4	13.8	9.8	2.6
	MTU1010	100	103	12	25	14.5	20.1	14.6	11.8	2.9
	R. Basmati	92	101	11	18	13.6	19.1	16.3	13.5	2.3
	DS20	98.1	107.4	13	22.4	12.8	16.7	13.4	10.7	3.1
	MTU1010	103	109	12	23	13.8	17.2	10.9	8.1	1.1
	R. Basmati	100	101	14	22	15.5	15.2	12.6	10.4	2.2
Variance	WS17	60.9	247.1	4.5	5	108.6	27.7	25.1	2.4	1.9
	DS18	236.7	327.8	16.9	5	48.4	16.7	15.9	3.4	0.9
	WS20	32.3	185.4	3.2	4.4	39	16.7	17.3	2.1	0.7
	DS20	62.4	67.6	7.4	4.8	11.6	13.3	13	1.2	2.4
Std Error (Dev)	WS17	7.8	15.7	2.1	2.2	10.4	5.3	5	1.6	1.4
	DS18	15.4	18.1	4.1	2.2	7	4.1	4	1.8	0.9
	WS20	5.7	13.6	1.8	2.1	6.2	4.1	4.2	1.5	0.8
	DS20	7.9	8.2	2.7	2.2	3.4	3.7	3.6	1.1	1.6
Skewness	WS17	0.4	−0.4	0.6	−0.3	0.6	0.6	0.7	0.3	2.4
	DS18	−0.4	−1.6	0.2	−0.2	0.8	1.1	1	−0.9	0.9
	WS20	0.5	−0.1	0.2	−0.8	1.2	1.2	1.9	0.6	1
	DS20	0.4	0.9	0.2	0	1	0.2	0.3	0.4	5.5
Kurtosis	WS17	−1	−0.8	0.8	0.9	−0.3	−0.4	0	−0.5	9.5
	DS18	1.3	9.5	−0.1	0.2	1.2	3.4	2.8	3.3	1.1
	WS20	0.8	−1	−0.3	1.7	2.9	2.9	7.3	−0.2	1
	DS20	−0.3	0.5	0.2	−0.2	2.4	1.5	1.4	1	40.3
Minimum	WS17	92	89	7	19.7	6.5	12.4	6.8	6.4	1
	DS18	37	8	7	16	3.6	8.5	5.3	1.2	0.4
	WS20	92	92	7	18	7.5	7.2	5.6	7.4	1.4
	DS20	80	92	7	17	6.2	7.3	4.5	8.1	1.1
Maximum	WS17	123	157	18	33	51	37.8	31.7	13.2	10.6
	DS18	131	153	27	27	42.9	35.2	30.5	13.3	5.1
	WS20	125	147	16	30	45	32.9	35.6	13.5	5.7
	DS20	118	135	22	27	25.9	28.4	26.3	14.3	15.5
Range	WS17	31	68	11	13.3	44.5	25.4	24.9	6.8	9.6
	DS18	94	145	20	11	39.3	26.7	25.2	12.1	4.7
	WS20	33	55	9	12	37.5	25.7	30	6.1	4.3
	DS20	38	43	15	10	19.7	21.1	21.8	6.2	14.4
Standard Error	WS17	0.8	1.6	0.22	0.23	1.08	0.54	0.52	0.16	0.14
	DS18	1.6	1.9	0.42	0.23	0.72	0.42	0.41	0.19	0.1
	WS20	0.6	1.4	0.19	0.21	0.64	0.42	0.43	0.15	0.09
	DS20	0.8	0.8	0.28	0.22	0.35	0.38	0.37	0.11	0.16
CV %	WS17	1.6	3.2	0.43	0.45	2.13	1.08	1.03	0.32	0.28
	DS18	3.2	3.7	0.84	0.45	1.42	0.84	0.82	0.38	0.19
	WS20	1.2	2.8	0.37	0.42	1.28	0.84	0.85	0.3	0.17
	DS20	1.6	1.7	0.55	0.44	0.7	0.75	0.74	0.23	0.32

WS: Wet Season; DS: Dry Season; DFF: Days to 50% Flowering (days); PH: Plant Height (cm); NT: No. of Tillers; PL: Panicle Length (cm); SPY: Single Plant Yield (g); ZnBR: Zinc Content in Brown Rice; ZnPR (ppm): Zinc Content in Polished Rice (ppm); FeBR: Iron Content in Brown Rice (ppm); FePR: Iron Content in Polished Rice (ppm).

**Table 2 plants-12-00144-t002:** Pearson correlation analysis for agronomic, yield-, and mineral-related traits evaluated during two wet seasons (WS) and two dry seasons (DS) (average).

	DFF	PH	NT	PL	SPY	ZnBR	ZnPR	FeBR	FePR
**DFF**	1.00	−0.01	0.06	0.02	0.28 **	0.04	0.02	−0.14	0.06
	1.00	−0.09	−0.12	0.03	−0.01	−0.06	−0.06	−0.35 ***	−0.24 *
**PH**	−0.01	1.00	0.10	0.25 *	0.04	−0.04	0.03	−0.04	−0.08
	−0.09	1.00	0.25 *	0.18	−0.20 *	0.00	0.05	0.11	−0.05
**NT**	0.06	0.10	1.00	−0.30 **	0.02	−0.06	−0.19	0.12	0.03
	−0.12	0.25 *	1.00	0.00	−0.01	−0.04	0.04	0.18	0.13
**PL**	0.02	0.25 *	−0.30 **	1.00	0.06	0.02	0.15	−0.26 *	−0.04
	0.03	0.18	0.00	1.00	−0.16	−0.04	−0.07	−0.02	0.03
**SPY**	0.28 **	0.04	0.02	0.06	1.00	−0.26 *	−0.21 **	−0.27 **	−0.19
	−0.01	−0.20 *	−0.01	−0.16	1.00	−0.17	−0.23 *	−0.09	−0.27 *
**ZnBR**	0.04	−0.04	−0.06	0.02	−0.26 *	1.00	0.79 ***	0.64 ***	0.40 ***
	−0.06	0.00	−0.04	−0.04	−0.17	1.00	0.76 ***	0.59 ***	0.23 *
**ZnPR**	0.02	0.03	−0.19	0.15	−0.21 **	0.79 ***	1.00	0.45 ***	0.35 ***
	−0.06	0.05	0.04	−0.07	−0.23 *	0.76 ***	1.00	0.47 ***	0.43 ***
**FeBR**	−0.14	−0.04	0.12	−0.26 *	−0.27	0.64 ***	0.45 ***	1.00	0.34 ***
	−0.35 ***	0.11	0.18	−0.02	−0.09	0.59 ***	0.47 ***	1.00	0.10
**FePR**	0.06	−0.08	0.03	−0.04	−0.19	0.40 ***	0.35 ***	0.34 ***	1.00
	−0.24 *	−0.05	0.13	0.03	−0.27 *	0.23 *	0.43 ***	0.10	1.00

*** *p* < 0.001; ** *p* < 0.01; * *p* < 0.05; DFF: Days to 50% Flowering; PH: Plant Height; NT: No. of Tillers; PL: Panicle Length; SPY: Single Plant Yield; ZnBR: Zinc Content in Brown Rice; ZnPR; Zinc Content in Polished Rice; FeBR: Iron Content in Brown Rice; FePR: Iron Content in Polished Rice. Upper value indicates wet season (WS), lower value indicates dry season (DS).

**Table 3 plants-12-00144-t003:** Importance of components.

	PC1	PC2	PC3	PC4	PC5	PC6	PC7	PC8	PC9
**Standard Deviation**	1.79	1.20	1.10	0.98	0.91	0.79	0.68	0.48	0.23
**Proportion of Variance**	0.35	0.16	0.13	0.11	0.09	0.07	0.05	0.03	0.01
**Cumulative Proportion**	0.35	0.52	0.65	0.76	0.85	0.92	0.97	0.99	1.00

**Table 4 plants-12-00144-t004:** Distribution of markers on the 12 linkage groups (LGs) of rice genetic map for the MTU1010 X Ranbir Basmati RIL population.

Linkage Grp	No. of SNPs Mapped	Map Distance (cM)	Density (SNPs/cM)
LG01	215	200	1.08
LG02	171	371	0.46
LG03	252	194	1.30
LG04	28	122	0.23
LG05	261	238	1.10
LG06	418	272	1.54
LG07	205	186	1.10
LG08	266	154	1.73
LG09	132	265	0.50
LG10	315	155	2.03
LG11	194	147	1.32
LG12	289	141	2.05
**Total**	**2746**	**2445**	**1.2**

**Table 5 plants-12-00144-t005:** Summary of major and minor QTLs for mineral traits (zinc (Zn) and iron (Fe)).

S. No.	Trait Name	QTL Name	Chr	Marker Interval	LOD	PVE (%)	Add	Allele
1	ZnBR-WS17	*qZnBR.2.1*	2	SNP_25682947-24248265	4.53	20.8	−2.56	P2
2	ZnBR-WS20	*qZnBR.4.1*	4	SNP_34361396-34862789	3.84	16.01	−1.81	P2
3	ZnBR-DS18	*qZnBR.5.1*	5	SNP_23987433-23987440	2.59	23.49	−1.3	P2
4	ZnBR-WS20	*qZnBR.9.1*	9	SNP_3990952-4052267	2.72	11.4	1.65	P1
5	ZnBR-DS20	*qZnBR.9.2*	9	SNP_3036200-9387676	2.55	13.88	−1.43	P2
6	ZnPR-WS17	*qZnPR.2.1*	2	SNP_25682947-24248265	4.75	19.42	−2.45	P2
7	ZnPR-WS20	*qZnPR.2.2*	2	SNP_35521280-24248265	3.46	11.78	−8.29	P2
8	ZnPR-WS17	*qZnPR.6.1*	6	SNP_28083826-2824031	2.81	12.34	−3.04	P2
9	ZnPR-DS18	*qZnPR.6.2*	6	SNP_28684145-30895032	3.19	8.33	−2.19	P2
10	ZnPR-DS18	*qZnPR.7.1*	7	SNP_14229996-13495315	3.31	8.1	3.52	P1
11	FeBR-WS17	*qFeBR.2.1*	2	SNP_24248265-10686645	4.11	15.73	−0.61	P2
12	FeBR-WS17	*qFeBR.3.1*	3	SNP_31758461-31908509	4.06	16.78	−0.85	P2
13	FeBR-WS17	*qFeBR.6.1*	6	SNP_2824031-28033555	2.73	11.27	−0.76	P2
14	FeBR-WS20	*qFeBR.5.1*	5	SNP_25431969-26111360	8.32	34.15	−0.95	P2
15	FePR-WS17	*qFePR.1.1*	1	SNP_12900648-35973332	10.38	28.73	−4.04	P2
16	FePR-DS18	*qFePR.2.1*	2	SNP_25682947-24248265	2.73	3.52	−0.27	P2
17	FePR-DS20	*qFePR.3.1*	3	SNP_13716046-28278891	26.59	36	−5.81	P2
18	FePR-WS17	*qFePR.5.1*	5	SNP_14288078-24423261	4.56	11.97	−0.68	P2
19	FePR-WS20	*qFePR.5.2*	5	SNP_28525048-27859206	2.91	8.68	0.69	P1
20	FePR-WS17	*qFePR.6.1*	6	SNP_2824031-28033555	2.81	12.43	−0.86	P2
21	FePR-DS18	*qFePR.8.1*	8	SNP_8033030-10615027	11.52	18.27	−0.81	P2
22	FePR-DS18	*qFePR.10.1*	10	SNP_18122916-17994542	4.12	5.53	0.37	P1
23	FePR-DS18	*qFePR.12.1*	12	SNP_214104199-217575057	13.17	21.55	−0.64	P2

Chr: Chromosome; QTL: Quantitative Trait Loci; cM: Centimorgan; PVE: Phenotypic Variance Explained; LOD: Logarithm of Odds; Add: Additive Effect; P1: Parent 1 (MTU1010); P2: Parent 2 (Ranbir Basmati); ZnBR: Zinc Content in Brown Rice; ZnPR: Zinc Content in Polished Rice; FeBR: Iron Content in Brown Rice; FePR: Iron Content in Polished Rice; WS: Wet Season; DS: Dry Season.

**Table 6 plants-12-00144-t006:** Summary of major and minor QTLs for agronomic and yield related traits.

S. No.	Trait	QTL Name	Chr	Marker Interval	LOD	PVE (%)	Add	Allele
1	DFF-DS18	*qDFF.1.1*	1	SNP_24168773-4543601	8.31	8.86	−8.33	P2
2	DFF-WS17	*qDFF.2.1*	2	SNP_12479844-31358768	4.17	16.05	4	P1
3	DFF-DS18	*qDFF.2.2*	2	SNP_18086957- 17352707	9.9	10.32	5.52	P1
4	DFF-DS20	*qDFF.2.3*	2	SNP_18086957-17352707	2.82	12.25	2.46	P1
5	DFF-DS20	*qDFF.2.4*	2	SNP_24248265-10686645	2.61	14.31	2.97	P1
6	DFF-DS18	*qDFF.2.5*	2	SNP_24248265-10686645	11.61	14.4	7.07	P1
7	DFF-DS18	*qDFF.3.1*	3	SNP_22060292-22880259	8.86	9.43	−6.4	P2
8	DFF-WS17	*qDFF.6.1*	6	SNP_1521959-2211457	3.04	6.63	−2.98	P2
9	DFF-WS17	*qDFF.6.2*	6	SNP_6569100-12310577	7.26	21.35	−4.68	P2
10	DFF-DS18	*qDFF.8.1*	8	SNP_14611920-15329887	16.54	20.96	13.22	P1
11	DFF-WS20	*qDFF.11.1*	11	SNP_11928941-1421311	3.68	19.25	−3.6	P2
12	PL-WS17	*qPL.2.1*	2	SNP_13823910-15278368	2.56	13.29	−0.73	P2
13	PL-DS20	*qPL.9.1*	9	SNP_18030272-17930222	3.3	8.03	0.83	P1
14	PL-DS18	*qPL.11.1*	11	SNP_116504977-12294027	2.65	12.39	1.47	P1
15	PL-DS20	*qPL.11.2*	11	SNP_125186904-123220554	5.51	16.51	−1.26	P2
16	PL-WS20	*qPL.12.1*	12	SNP_21721652-21797699	3.2	15.88	0.83	P1
17	SPY-DS18	*qSPY.3.1*	3	SNP_32176877-962276	2.63	11.26	4.71	P1
18	SPY-WS17	*qSPY.6.1*	6	SNP_22324866-1691751	3.57	15.32	−4.98	P2
19	SPY-WS17	*qSPY.9.1*	9	SNP_18053950-18053949	2.82	11.28	3.74	P1
20	SPY-DS20	*qSPY.9.2*	9	SNP_18915073-18745200	2.69	12.99	1.23	P1
21	NT-DS20	*qNT.6.1*	6	SNP_14518832-15621571	2.51	12.03	1.59	P1
22	NT-WS17	*qNT.9.1*	9	SNP_17649556-17654662	3.24	17.94	−0.81	P2
23	NT-WS20	*qNT.9.2*	9	SNP_17887929-18744623	3.33	10.37	−0.73	P2
24	NT-WS20	*qNT.10.1*	10	SNP_22330346-18248511	2.57	11.02	1.04	P1

Chr: Chromosome; QTL: Quantitative Trait Loci; cM: Centimorgan; PVE: Phenotypic Variance Explained; LOD: Logarithm Of Odds; Add: Additive Effect; P1: Parent 1 (MTU1010); P2: Parent 2 (Ranbir Basmati); DFF: Days to 50% Flowering; PL: Panicle Length; SPY: Single Plant Yield; NT: No. of Tillers; WS: Wet Season; DS: Dry Season.

## Data Availability

Not applicable.

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
