# Peer review of "Identification of Genomic Regions Associated with High Grain Zn Content in Polished Rice Using Genotyping-by-Sequencing (GBS)"

_plants, 2022, doi:10.3390/plants12010144_

Round 1

Reviewer 1 Report (Previous Reviewer 1)

My first round big doubt was about the experimental design and now the authors mentioned that they used RCBD. I should expect ANOVA from this design and authors should add before accepting the manuscript for publication.

Author Response

Reviewer 2 Report (Previous Reviewer 2)

In this paper, Fe and Zn micronutrient QTLs in rice have been identified by using GBS generated markers on RIL's resulting from  a cross of high and low scoring parents. The study is well presented and should be of high interest to other readers even beyond this field. While the language has improved from the first version, there still remain some errors that should be corrected though, mostly with incomplete/ incorrect sentences.

L12 is important step to

is an important step

L13 to develop the biofortified rice varieties

to develop biofortified rice varieties

L16 spanning genetic distance

spanning a genetic distance

L41 many enzymes have great importance

many enzymes which have great importance

L68 the yield through agronomic, conventional

the yield achieved through agronomic, conventional

L77 (QTLs) for high Zn in grains would help in

(QTLs) for high Zn in grains which would help in

L88 making them ideal choice for

making them the ideal choice for

L98 are some of the genome sequencing methods that have been developed

[remove, it is doubled]

L103 using recombinant inbred line (RIL) population derived from MTU1010 x Ranbir Basmati cross

using a recombinant inbred line (RIL) population derived from a MTU1010 x Ranbir Basmati cross

L241 Five major affect QTLs

Five major effect QTLs

L244 Majority of QTLs showed increase additive effect indicating that the allele contributed was from donor parent Ranbir Basmati.

[sentence!]

L253  was identified and the allele were from Ranbir

were identified and the alleles were from Ranbir

L254 One QTL each on Chr02...

[this is not a complete sentence]

L258 Nine QTLs were identified for FePR trait

Nine QTLs were identified for the FePR trait

L342 One epistatic interaction each for FeBR on Chr04 was interacted with Chr05 with PVE 16.16% and FePR on Chr02 interacted with Chr10 with PVE 26.83%.

[sentence!]

L385 The donor parent Ranbir Basmati noted higher expression of these 386 two genes than the two RILs having Zn content lesser than donor parent.

[sentence! Ranbir Basmati would not note anything]

L390 Biofortification, considered as long-term sustainable strategies

Biofortification is considered as long-term sustainable strategies

L400 eliminated by employing. NGS generated markers

eliminated by employing NGS generated markers

L403 in cost effective manner

in a cost effective manner

L408 using GBS approach

using a GBS approach

L431 with >24 ppm

[of what?]

Author Response

Thank you

Reviewer 3 Report (Previous Reviewer 4)

no more question

Author Response

Thank you for your comments.

This manuscript is a resubmission of an earlier submission. The following is a list of the peer review reports and author responses from that submission.

Round 1

Reviewer 1 Report

Intro

L54-56. The sentence starting "According..." should need rephrasing

L61,L62. Is it "trail" or "trial"?

L77.  "Quantitative Trait Loci (QTL)" - quantitative trait loci (QTL)

L80. phenotyping - phenotypic

L95-98. Sentence could rephrase for better clarity

Materials and Methods

L501. The field experiment lacked mentioning the experimental design. 

L501. phenotype data - phenotypic data

L509. Details on the phenology- and yield-related traits collection should be provided. For instance, how was grain yield estimated? from the whole plot? only from some plants/plant?

L526. For which of analysis did the GGE Biplot bio-tools, gge, agricolae, Fac-526 toMinerR and factoextra packages used? it should be detailed.

L556. MAF 0.05 should be changed to MAF > 0.05 

L557. How could TASSEL be used as an in-house script? It is a publicly available standalone software package/

L561-562. The sentence needs rephrasing, otherwise is confusing

******** I felt the experimental design should be well elaborated and should be acceptable***********

Reviewer 2 Report

The study describes the identification of QTL markers for Zn and Fe content from a rice RIL. While the basic findings are sound, the study would have much benefitted from inclusion of at least a second location, as Zn content is known to vary considerably depending on the environment (eg Rao et al 2020, Frontiers in nutrition Vol 7, article 26). Additionally, the measuring of just five plants for phenotypic data seems borderline sufficient for following calculations of mean, standard error, skewness and so forth. For publication, a minimum required would be improvement of the language. While materials and methods is largely well written, there are a number of errors,  imprecise use of language, sentence constructions that need to be corrected. I add examples below, but they are not exhaustive. The whole  manuscript will need careful re-reading and editing for correct and exact language.

line 40 Microelements, such as iron (Fe), zinc (Zn), copper (Cu) and manganese (Mn) are found in many enzymes and has great importance

Microelements, ... have great importance

line 42 Zn serves as a major co-factor for more than 300 enzymes, 2000  transcription factors, and involves in the metabolism

Zn serves..., and is involved in the metabolism

line 51 as they depend on plant-based diet that have

as they depend on plant-based diets that have

line 60 The present international target as well as AICRIP biofortification trail is 28 ppm

The present international target as well as that from the AICRIP biofortification trial is 28 ppm

line 61 Through AICRIP rice biofortification trail

Through the AICRIP rice biofortification trial

line 75 and influenced by environmental factors

and is influenced by environmental factors

line 86 wide occurrence across the genome (~100 – 500 bp

wide occurrence across the genome (one marker every 100 – 500 bp

line 92 NGS technology based on genome reduction with restriction enzyme

NGS technology based on genome target reduction based on restriction enzyme use

line 102 present study targeted on constructing a high-density

present study is targeted on constructing a high-density

line 104 Potential genes underlying between the marker intervals of the QTLs identified

?

line 370 These two candidate genes or within the identified QTL region are being further investigated

? These two candidate genes within the identified QTL region are being further investigated

line 395 Markers assisted breeding has a vast potential

Marker assisted breeding has a vast potential

line 399 were successfully eliminated by employing next generation markers such as SNPs

were successfully eliminated by employing ngs generated markers such as SNPs

line 402 (GBS) technique has facilitated by identifying key genomic regions for both complex

(GBS) technique has facilitated the identification of key genomic regions for both complex

line 414 mineral content across wet and dry seasons (Kharif and Rabi)

proper citation, or is this personal communication?

line 420 reported positive relationship between Zn and Fe

? reported positive correlation between Zn and Fe

line 473 Epistasis should be underscored in studying complex traits because

? should be taken into account

line 479 major QTLs with grain mineral elements and/or show consistency

major QTLs invlved in grain mineral elements content which show consistency

line 542 These libraries were PCR amplified (8-12 cycles) using a PhusionTM polymerase kit (Fisher Scientific, United Kingdom) to remove excess adapters

[PCR is not used to remove adapters; I assume PCR has been done followed by a not mentioned purification step to remove excess adapters]

Reviewer 3 Report

"A recombinant inbred line (RIL) population derived

from MTU1010 (high yielding rice variety) and Ranbir Basmati (high Zn rice variety) was phenotyped for four seasons and genotyped by genotyping-by-sequencing (GBS)" - Add the number of RILs and rephrase the sentence.

"days to 50% flowering (qDFF.2.2 and qDFF.2.3)" - this can be removed and better to keep the focus on Fe and Zn.

Results

"In WS 17, G3 RIL showed Zn content above threshold value of 24 ppm as well as SPY  above 26 gm of yield check (IR64) (Figure 1). In WS 20, G5 RIL showed Zn content above threshold value (24ppm) and G3 along with another seven RILs noted SPY. In the DS 18, G1 noted Zn content noted above threshold value (24ppm) and nine lines noted SPY above the yield check (26gm). In DS 20, G6 noted Zn content above threshold value and G3 line noted on par yield with yield check. Based on mean Zn content and SPY of the four seasons none of the genotypes showed Zn content above threshold value whereas seven RILs showed SPY ≥ the yield check (26gm)." - what about Fe.

Distribution of both Fe and Zn is very less.

Better to combine Figure 1 and Figure 2 as a single multi-panel figure.

The major constraint for the study is a very less number of RILs. QTL mapping with 92 RILs is not at all reliable. Authors need at least 384 RILs for such a complex and phenotypicaly challenging traits.

Reviewer 4 Report

1. The phenotypic data of parents should be provided.

2. How many replicates were conducted for field experiments in each season?

3. It's better to merge Table 2 into a single table.

4. Why the marker number on each chromosome listed in Table 4 was not equal that illustrated in Figure 5?

5. No conclusion part in the MS.

6. The method to measure Zn and Fe should be detailed. What's the accuracy of ED-XRF to measure Zn or Fe content?